# Towards Robustness against Unsuspicious Adversarial Examples

## Abstract

Despite the remarkable success of deep neural networks, significant concerns have emerged about their robustness to adversarial perturbations to inputs. While most attacks aim to ensure that these are imperceptible, *physical* perturbation attacks typically aim for being unsuspicious, even if perceptible. However, there is no universal notion of what it means for adversarial examples to be unsuspicious. We propose an approach for modeling suspiciousness by leveraging cognitive salience. Specifically, we split an image into foreground (salient region) and background (the rest), and allow significantly larger adversarial perturbations in the background, while ensuring that cognitive salience of background remains low. We describe how to compute the resulting non-salience-preserving dual-perturbation attacks on classifiers. We then experimentally demonstrate that our attacks indeed do not significantly change perceptual salience of the background, but are highly effective against classifiers robust to conventional attacks. Furthermore, we show that adversarial training with dual-perturbation attacks yields classifiers that are more robust to these than state-of-the-art robust learning approaches, and comparable in terms of robustness to conventional attacks.

## 1 Introduction

An observation by Szegedy et al. (2014) that state-of-the-art deep neural networks that exhibit exceptional performance in image classification are fragile in the face of small adversarial perturbations of inputs has received a great deal of attention. A series of approaches for designing adversarial examples followed (Szegedy et al., 2014; Goodfellow et al., 2015; Carlini & Wagner, 2017), along with methods for defending against them (Papernot et al., 2016b; Madry et al., 2018), and then new attacks that defeat prior defenses, and so on. Attacks can be roughly classified along three dimensions: 1) introducing small $l_p$-norm-bounded perturbations, with the goal of these being imperceptible to humans (Madry et al., 2018), 2) using non-$l_p$-based constraints that capture perceptibility (often called *semantic perturbations*) (Bhattad et al., 2020), and 3) modifying physical objects, such as stop signs (Eykholt et al., 2018), in a way that does not arouse suspicion. One of the most common motivations for the study of adversarial examples is safety and security, such as the potential for attackers to compromise the safety of autonomous vehicles that rely on computer vision (Eykholt et al., 2018). However, while imperceptibility is certainly sufficient for perturbations to be unsuspicious, it is far from necessary, as physical attacks demonstrate. On the other hand, while there are numerous formal definitions that capture whether noise is perceptible (Moosavi-Dezfooli et al., 2016; Carlini & Wagner, 2017), what makes adversarial examples suspicious has been largely informal and subjective.

We propose a simple formalization of an important aspect of what makes adversarial perturbations unsuspicious. Specifically, we make a distinction between image foreground and background, allowing significantly more noise in the background than the foreground. This idea stems from the notion of cognitive salience (Borji et al., 2015; Kmmerer et al., 2017; He & Pugeault, 2018), whereby an image can be partitioned into the two respective regions to reflect how much attention a human viewer pays to the different parts of the captured scene. In effect, we posit that perturbations in the foreground, when visible, will arouse significantly more suspicion (by being cognitively more salient) than perturbations made in the background.

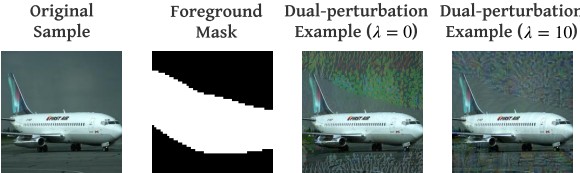

Figure 1: An illustration of dual-perturbation attacks. Adversarial examples are with large $\ell_\infty$ perturbations on the background ($\epsilon_B = 20/255$) and small $\ell_\infty$ perturbations on the foreground ($\epsilon_F = 4/255$). A parameter $\lambda$ is used to control background salience explicitly. A larger $\lambda$ results in less salient background under the same magnititude of perturbation.

Our first contribution is a formal model of such *dual-perturbation attacks*, which is a generalization of the $l_p$-norm-bounded attack models (see, e.g., Figure 1), but explicitly aims to ensure that adversarial perturbation does not make the background highly salient. Second, we propose an algorithm for finding adversarial examples using this model, which is an adaptation of the PGD attack (Madry et al., 2018). Third, we present a method for defending against dual-perturbation attacks based on the adversarial training framework (Madry et al., 2018). Finally, we present an extensive experimental study that demonstrates that (a) the proposed attacks are significantly stronger than PGD, successfully defeating all state-of-the-art defenses, (b) proposed defenses using our attack model significantly outperform state-of-the-art alternatives, *with relatively small performance degradation on non-adversarial instances*, and (c) proposed defenses are comparable to, or better than alternatives *even against traditional attacks*, such as PGD.

**Related Work:** Recent studies have shown that neural networks are vulnerable to adversarial examples. A variety of approaches have been proposed to produce adversarial examples (Szegedy et al., 2014; Goodfellow et al., 2015; Papernot et al., 2016a; Moosavi-Dezfooli et al., 2016; Carlini & Wagner, 2017). These approaches commonly generate adversarial perturbations within a bounded $\ell_p$ norm so that the perturbations are imperceptible. A related thread has considered the problem of generating adversarial examples that are semantically imperceptible without being small in norm (Brown et al., 2018; Bhattad et al., 2020), for example, through small perturbations to the color scheme. However, none of these account for the perceptual distinction between the foreground and background of images.

Numerous approaches have been proposed for defending neural networks against adversarial examples (Papernot et al., 2016b; Carlini & Wagner, 2017; Madry et al., 2018; Cohen et al., 2019; Madry et al., 2018; Raghunathan et al., 2018). Predominantly, these use $\ell_p$-bounded perturbations as the threat model, and while some account for semantic perturbations (e.g. Mohapatra et al. (2020)), none consider perceptually important difference in suspiciousness between foreground and background.

Two recent approaches by Vaishnavi et al. (2019) and Brama & Grinshpoun (2020) have the strongest conceptual connection to our work. Both are defense-focused by either eliminating (Vaishnavi et al., 2019) or blurring (Brama & Grinshpoun, 2020) the background region for robustness. However, they *assume* that we can reliably segment an image *at prediction time*, leaving the approach vulnerable to attacks on image segmentation (Arnab et al., 2018). Xiao et al. (2020) propose to disentangle foreground and background signals on images but unsuspiciousness of their attacks is not ensured.

## 2 BACKGROUND

### 2.1 ADVERSARIAL EXAMPLES AND ATTACKS

The problem of generating adversarial examples is commonly modeled as follows. We are given a a learned model $h_\theta(\cdot)$ parameterized by $\theta$ which maps an input $x$ to a $k$-dimensional prediction, where $k$ is the number of classes being predicted. The final predicted class $y_p$ is obtained by $y_p = \arg\max_i h_\theta(x)_i$, where $h_\theta(x)_i$ is the $i$th element of $h_\theta(x)$. Now, consider an input $x$ along with a correct label $y$. The problem of identifying an adversarial example for $x$ can be captured by

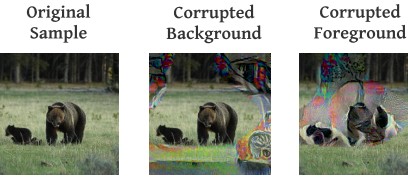

Figure 2: Semantic distinction between foreground and background. Left: Original image of bears. Middle: Adversarial example with $\ell_\infty$ bounded perturbations ($\epsilon = 40/255$) on the background, the sematic meaning (bear) is preserved. Right: Adversarial example with $\ell_\infty$ bounded perturbations ($\epsilon = 40/255$) on the foreground, with more ambiguous semantics.

the following optimization problem:

$$\max_{\boldsymbol{\delta} \in \Delta(\epsilon)} \mathcal{L}\left(h_{\boldsymbol{\theta}}(\boldsymbol{x} + \boldsymbol{\delta}), y\right), \tag{1}$$

where $\mathcal{L}(\cdot)$ is the adversary's utility function (for example, the loss function used to train the classifier $h_{\boldsymbol{\theta}}$). $\Delta(\epsilon)$ is the feasible perturbation space which is commonly represented as a $\ell_p$ ball: $\Delta(\epsilon) = \{\boldsymbol{\delta} : \|\boldsymbol{\delta}\|_p \leq \epsilon\}$.

A number of approaches have been proposed to solve the optimization problem shown in Eq. (1), among which two are viewed as state of the art: *CW attack* developed by Carlini & Wagner (2017), and *Projected Gradient Descent (PGD) attack* proposed in Madry et al. (2018). In this work, we focus on the PGD attack with $\ell_\infty$ and $\ell_2$ as the distance metrics.

### 2.2 ROBUST LEARNING

An important defense approach that has proved empirically effective even against adaptive attacks is *adversarial training* (Szegedy et al., 2014; Cohen et al., 2019; Goodfellow et al., 2015; Madry et al., 2018). The basic idea of adversarial training is to produce adversarial examples and incorporate these into the training process. Formally, adversarial training aims to solve the following robust learning problem:

$$\min_{\boldsymbol{\theta}} \frac{1}{|D|} \sum_{\boldsymbol{x},y \in D} \max_{\|\boldsymbol{\delta}\|_p \leq \epsilon} \mathcal{L}\left(h_{\boldsymbol{\theta}}(\boldsymbol{x} + \boldsymbol{\delta}), y\right), \tag{2}$$

where $D$ is the training dataset. In practice, this problem is commonly solved by iteratively using the following two steps (Madry et al., 2018): 1) use a PGD (or other) attack to produce adversarial examples of the training data; 2) use any optimizer to minimize the loss of those adversarial examples. It has been shown that adversarial training can significantly boost the adversarial robustness of a classifier against $\ell_p$ attacks, and it can be scaled to neural networks with complex architectures.

## 3 DUAL-PERTURBATION ATTACKS

### 3.1 MOTIVATION

Our threat model is motivated by the *feature integration theory* (Treisman & Gelade, 1980) in cognitive science: regions that have features that are different from their surroundings are more likely to catch a viewer's gaze. Such regions are called *salient regions*, or *foreground*, while the others are called *background*. Accordingly, for a given image, the semantics of the object of interest is more likely to be preserved in the foreground, as it catches more visual attention of a viewer compared to the background. If the foreground of an image is corrupted, then the semantics of the object of interest is broken. In contrast, the same extent of corruption in the background nevertheless preserves the overall semantic meaning of the scene captured (see, e.g., Figure 2). Indeed, detection of salient regions, as well as the segmentation of foreground and background, have been extensively studied in computer vision (Borji et al., 2015). These approaches either predict human fixations, which are sparse bubble-like salient regions sampled from a distribution (Kmmerer et al., 2017), or salient objects that contain smooth connected areas in an image (He & Pugeault, 2018).

Despite this important cognitive distinction between foreground and background, essentially all of the attacks on deep neural networks for image classification make no such distinction, even though a number of other semantic factors have been considered (Bhattad et al., 2020; Mohapatra et al., 2020). Rather, much of the focus has been on adversarial perturbations that are *not noticeable* to a human, but which are applied equally *to the entire image*. However, in security applications, the important issue is not merely that an attack cannot be noticed, but that whatever observed is *not suspicious*. This is, indeed, the frame of reference for many high-profile *physical* attacks on image classification, which are clearly visible, but not suspicious because they hide in the "human psyche", that is, are easily ignored (Sharif et al., 2016; Eykholt et al., 2018). The main goal of the threat model we introduce next is therefore to capture more precisely the notion that an adversarial example is not suspicious by leveraging the cognitive distinction between foreground and background of an image.

## 3.2 DUAL-PERTURBATION ATTACKS

At the high level, our proposed threat model involves producing small (imperceptible) adversarial perturbations in the foreground of an image, and larger perturbations in the background. This can be done by incorporating state-of-the-art attacks into our method: we can use one attack with small $\epsilon$ in the foreground, and another with a large $\epsilon$ in the background. Consequently, we term our approach *dual-perturbation attacks*. Note that these clearly generalize the standard small-norm (e.g., PGD) attacks, since we can set the $\epsilon$ to be identical in both the foreground and background. However, the key consideration is that after we add the large amount of noise to the background, *we must ensure that we do not thereby make it highly salient to the viewer*. We capture this second objective by including in the optimization problem a *salience* term that decreases with increasing salience of the background.

Formally, the *dual-perturbation* attack solves the following optimization problem:

$$\max_{||\boldsymbol{\delta} \circ \mathcal{F}(\boldsymbol{x})||_p \leq \epsilon_F, ||\boldsymbol{\delta} \circ \mathcal{B}(\boldsymbol{x})||_p \leq \epsilon_B} \mathcal{L}\left(h_{\boldsymbol{\theta}}(\boldsymbol{x} + \boldsymbol{\delta}), y\right) + \lambda \cdot \mathcal{S}\left(\boldsymbol{x} + \boldsymbol{\delta}\right), \tag{3}$$

where $\mathcal{S}(\boldsymbol{x} + \boldsymbol{\delta})$ measure the relative salience of the foreground compared to background after adversarial noise $\boldsymbol{\delta}$ has been added, with $\lambda$ a parameter that explicitly balances the two objectives: maximizing predicted loss on adversarial examples, and limiting background salience (compared to foreground) so that the adversarial example produced is unsuspicious. Here $\mathcal{F}$ returns the mask matrix constraining the area of the perturbation in the foreground, and $\mathcal{B}$ returns the mask matrix restricting the area of the perturbation in the background, for an input image $\boldsymbol{x}$. $\mathcal{F}(\boldsymbol{x})$ and $\mathcal{B}(\boldsymbol{x})$ have the same dimension as $\boldsymbol{x}$ and contain 1s in the area which can be perturbed and 0s elsewhere. $\circ$ denotes element-wise multiplication for matrices. Hence, we have $\boldsymbol{x} = \mathcal{F}(\boldsymbol{x}) + \mathcal{B}(\boldsymbol{x})$ which indicates that any input image can be decomposed into two independent images: one containing just the foreground, and the other containing the background.

We model the suspiciousness $\mathcal{S}(\boldsymbol{x})$ of an input image $\boldsymbol{x}$ by leveraging a recent computational model of image salience, DeepGaze II (Kmmerer et al., 2017). DeepGaze II outputs predicted pixel-level density of human fixations on an image with the total density over the entire image summing to 1. Our measure of relative salience of the foreground to background is the *foreground score*, which is defined as $\mathcal{S}(\boldsymbol{x}) = \sum_{i \in \{k | \mathcal{F}(\boldsymbol{x})_k \neq 0\}} s_i$, where $s_i$ is the saliency score produced by DeepGaze II for pixel $i$ of image $\boldsymbol{x}$. Since foreground, as a fraction of the image, tends to be around 50-60%, a score significantly higher than 0.5 indicates that predicted human fixation is relatively localized to the foreground.

A natural approach for solving the optimization problem shown in Equation 3 is to apply an iterative method, such as the PGD attack. However, the use of this approach poses two challenges in our setting. First, as in the PGD attack, the problem is non-convex, and PGD only converges to a local optimum. We can address this issue by using *random starts*, i.e., by randomly initializing the starting point of the adversarial perturbations, as in Madry et al. (2018). Second, and unlike PGD, the optimization problem in Equation 3 involves *two hard constraints* $||\boldsymbol{\delta} \circ \mathcal{F}(\boldsymbol{x})||_p \leq \epsilon_F$ and $||\boldsymbol{\delta} \circ \mathcal{B}(\boldsymbol{x})||_p \leq \epsilon_B$. Thus, the feasible region of the adversarial perturbation $\boldsymbol{\delta}$ is not an $\ell_p$ ball, which makes computing the projection $\mathcal{P}_\epsilon$ computationally challenging in high-dimensional settings. To address this challenge, we split the *dual-perturbation* attack into two individual processes in each iteration, one for the adversarial perturbation in the foreground and the other for the background, and then merge these two perturbations when computing the gradients, like a standard PGD attack. Full details of our algorithms for computing dual perturbation examples are provided in Appendix A.

Now, the question that remains is how to partition an input image $\boldsymbol{x}$ into foreground, $\mathcal{F}(\boldsymbol{x})$, and background, $\mathcal{B}(\boldsymbol{x})$. We address this next.

### 3.3  IDENTIFYING FOREGROUND AND BACKGROUND

Given an input $\boldsymbol{x}$, we aim to compute $\mathcal{F}(\boldsymbol{x})$, the foreground mask and $\mathcal{B}(\boldsymbol{x})$, the background mask. We consider two approaches for this: fixation prediction and segmentation.

Our first method leverages the fixation prediction approach (Kmmerer et al., 2017) to identify foreground and background. This enables a general approach for foreground-background partition as fixation predictions are not limited to any specific collection of objects. Specifically, we first use DeepGaze II (Kmmerer et al., 2017) to output predicted pixel-level density of human fixations on an image. We then divide the image into foreground and background by setting a threshold $t = 0.5 \cdot (s_{min}(\boldsymbol{x}) + s_{max}(\boldsymbol{x}))$ for each input image $\boldsymbol{x}$ where $(s_{min}, s_{max})$ are the minimum and maximum values of human fixation on pixels of $\boldsymbol{x}$. Pixels with larger values than $t$ are grouped into the foreground, and the others are identified as background subsequently.

Our second approach is to make use of semantic segmentation to provide a partition of the foreground and background in pixel level. This can be done in two steps: First, we use state-of-the-art paradigms for semantic segmentation (e.g., Long et al. (2015)) to identify pixels that belong to each corresponding object, as there might be multiple objects in an image. Next, we identify the pixels that belong to the object of interest as the foreground pixels, and the others as background pixels.

We use both of the above approaches in dual-perturbation attacks when evaluating the robustness of classifiers, as well as designing robust models. More details are available in Section 5.

## 4  DEFENSE AGAINST DUAL-PERTURBATION ATTACKS

Once we are able to compute the dual-perturbation attack, we can incorporate it into conventional adversarial training paradigms for defense, as it has been demonstrated that adversarial training is highly effective in designing classification models that are robust to a given attack. Specifically, we replace the PGD attack in the adversarial training framework proposed by Madry et al. (2018), with the proposed dual-perturbation attack. We term this approach *AT-Dual*, which aims to solve the following optimization problem:

$$\min_{\boldsymbol{\theta}} \frac{1}{|D|} \sum_{\boldsymbol{x},y \in D} \max_{\substack{||\boldsymbol{\delta} \circ \mathcal{F}(\boldsymbol{x})||_p \leq \epsilon_F, \\ ||\boldsymbol{\delta} \circ \mathcal{B}(\boldsymbol{x})||_p \leq \epsilon_B}} \mathcal{L}\left(h_{\boldsymbol{\theta}}(\boldsymbol{x} + \boldsymbol{\delta}), y\right) + \lambda \cdot \mathcal{S}\left(\boldsymbol{x} + \boldsymbol{\delta}\right). \tag{4}$$

Note that *AT-Dual* needs to identify background and foreground for any input when solving the inner maximization problems in Equation 4 at training time. At prediction time, our approaches classify test samples like any standard classifiers, which is independent of the semantic partitions so as to close the backdoors to attacks on object detection approaches (Xie et al., 2017). We evaluate the effectiveness of our approaches in Section 5.

## 5  EXPERIMENTS

### 5.1  EXPERIMENTAL SETUP

**Datasets**. We conducted the experiments on the following three datasets (detailed in Appendix B): The first is Segment-6 (Cong & Prakash, 2019), which are images with $32 \times 32$ pixels obtained by pre-processing the Microsoft COCO dataset (Lin et al., 2014) to make it compatible with image classification tasks. We directly used the semantic segmentation based foreground masks provided in this dataset. Our second dataset is STL-10, a subset that contains images with $96 \times 96$ pixels. Our third dataset is ImageNet-10, a 10-class subset of the ImageNet dataset (Deng et al., 2009). We cropped all its images to be with $224 \times 224$ pixels. For STL-10 and ImageNet-10, we used fixation prediction to identify foreground and background as described in Section 3.

**Baselines**. We consider *PGD* attack as a baseline adversarial model, and *Adversarial Training with PGD Attacks* as a baseline robust classifier. We also consider a classifier trained on non-adversarial

data (henceforth, *Clean*). Additionally, we consider *Randomized Smoothing* (Cohen et al., 2019) and defer the corresponding results to Appendix J.

**Evaluation Metrics**. We use two standard evaluation metrics for both attacks and defenses: 1) accuracy of prediction on clean test data where no adversarial attacks were attempted. 2) adversarial accuracy, which is accuracy when adversarial inputs are used in place of clean inputs.

Throughout our evaluation, we used both $\ell_2$ and $\ell_\infty$ norms to measure the magnitude of added adversarial perturbations. Due to space limitations, we only present experimental results of the *Clean* model and classification models that are trained to be robust to $\ell_2$ norm attacks using the ImageNet-10 dataset. The results for $\ell_\infty$ norm and other datasets are similar and deferred to Appendix.

In the following experiments, all classifiers were trained with 20 epochs on a ResNet34 model (He et al., 2016) pre-trained on ImageNet and with a customized final fully connected layer. Specifically, we trained AT-PGD by using 50 steps of $\ell_2$ PGD attack with $\epsilon = 2.0$, and AT-Dual by using 50 steps of $\ell_2$ dual-perturbation attack with $\{\epsilon_F, \epsilon_B, \lambda\} = \{2.0, 20.0, 0.0\}$ at each training epoch. At test time, we used both $\ell_2$ PGD and dual-perturbation attacks with 100 steps to evaluate robustness.

## 5.2 SALIENCY ANALYSIS OF DUAL-PERTURBATION ADVERSARIAL EXAMPLES

We begin by considering a natural question: is our particular distinction between foreground and background actually consistent with cognitive salience? In fact, this gives rise to two distinct considerations: 1) whether foreground as we identify it is in fact significantly more salient than the background, and 2) if so, whether background becomes significantly more salient *as a result of our dual-perturbation attacks*. We answer both of these questions by appealing to DeepGaze II (Kmmerer et al., 2017) to compute the *foreground score (FS)* of dual-perturbation examples as described in Section 3, and using the accuracy of different classifiers on dual-perturbation examples with different background salience.

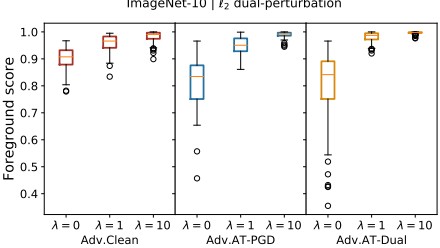
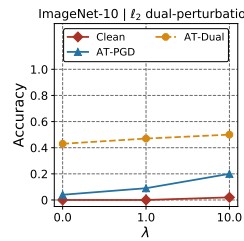

Figure 3: Saliency analysis. Dual-perturbation attacks are performed by using $\{\epsilon_F, \epsilon_B\} = \{2.0, 20.0\}$ and a variety of $\lambda$ displayed in the figure. Left: foreground scores of dual-perturbation examples in response to different classifiers. Right: accuracy of classifiers on dual-perturbation examples with salience control.

Figure 3 presents the answer to both of the questions above. First, observe that in Figure 3, *FS* (vertical axis) is typically well above 0.5, and in most cases above 0.9, for all attacks. Second, this is true whether we attack the *Clean* model, or either *AT-PGD* or *AT-Dual* robust models. Particularly noteworthy, however, is the impact that the parameter $\lambda$ has on the *FS*, especially when robust classifiers are employed. Recall that $\lambda$ reflects the relative importance of salience in generating adversarial examples, with larger values forcing our approach to pay more attention to preserving unsuspiciousness of background relative to foreground. As we increase $\lambda$, we note significantly higher *FS*, i.e., lower background salience (again, Figure 3, left). Figure 1 offers a visual illustration of this effect.

As significantly, Figure 3 (right) shows that moderately increasing $\lambda$ does not significantly reduce the effectiveness of the attack, on either the *Clean* or the robust classifiers.

## 5.3 DUAL-PERTURBATION ATTACKS ON ROBUST CLASSIFIERS

Next, we evaluate the effectiveness of dual-perturbation attacks against state-of-the-art robust learning methods, as well as the effectiveness of adversarial training that uses dual-perturbation attacks for

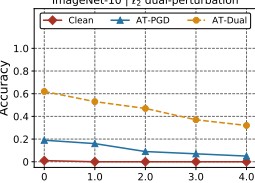 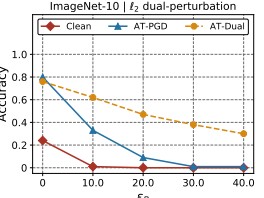 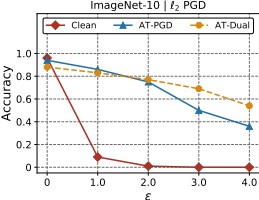

Figure 4: Robustness to white-box $\ell_2$ attacks on ImageNet-10. Left: dual-perturbation attacks with different foreground distortions. $\epsilon_B$ is fixed to be 20.0 and $\lambda = 1.0$. Middle:dual-perturbation attacks with different background distortions. $\epsilon_F$ is fixed to be 2.0 and $\lambda = 1.0$. Right: PGD attacks.

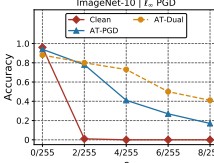 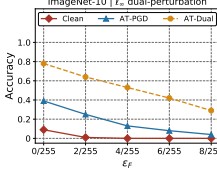 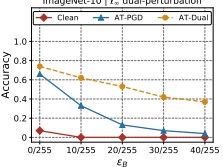 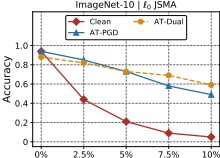

Figure 5: Robustness to additional white-box attacks on ImageNet-10. Left: 20 steps of $\ell_\infty$ PGD attacks. Middle left: 20 steps of $\ell_\infty$ dual-perturbation attacks with different foreground distortions. $\epsilon_B$ is fixed to be 20/255 and $\lambda = 1.0$. Middle right: 20 steps of $\ell_\infty$ dual-perturbation attacks with different background distortions. $\epsilon_F$ is fixed to be 4/255 and $\lambda = 1.0$. Right: $\ell_0$ JSMA attacks.

generating adversarial examples. We begin by considering white-box attacks, and subsequently evaluate transferability. Due to space limitations, we defer the results of transferability to Appendix D.

The results for white-box attacks are presented in Figure 4. First, consider the dual-perturbation attacks (left and middle plots). Note that in all cases these attacks are highly successful against the baseline robust classifier (AT-PGD); indeed, even relatively small levels of foreground noise yield near-zero accuracy when accompanied by sufficiently large background perturbations. For example, when the perturbation to the foreground is $\epsilon_F = 2.0$ and background perturbation is $\epsilon_B = 20.0$, *AT-PGD* achieves robust accuracy below $10\%$. In contrast, AT-Dual remains significantly more robust, with an improvement of up to $40\%$ compared to the baseline. Second, consider the standard PGD attacks (right plot). It can be observed that all of the robust models are successful against the $\ell_2$ PGD attacks. However, our defense exhibit moderately higher robustness than the baselines under large distortions of PGD attacks, without sacrificing much in accuracy on clean data. For example, when the perturbation of the $\ell_2$ PGD attack is above $\epsilon = 3.0$, *AT-Dual* can achieve 20% more accuracy.

## 5.4 GENERALIZABILITY OF DEFENSE

It has been observed that models robust against $l_p$-norm-bounded attacks for one value of $p$ can be fragile when facing attacks with a different norm $l_{p'}$ (Sharma & Chen, 2018). Here, our final goal is to present evidence that the approaches for defense based on dual-perturbation attacks remain relatively robust even when faced with attacks generated using different norms. Here, we show this when our models are trained using the $l_2$-bounded attacks, and evaluated against other attacks using other norms. The results are presented in Figure 5. We consider three alternative attacks: 1) PGD using the $l_\infty$-bounded perturbations, as in Madry et al. (2018) (left in Figure 5) 2) dual-perturbation attacks with $l_\infty$-norm bounds (middle left and middle rigt in Figure 5), and 3) JSMA, a $l_0$-bounded attack (Papernot et al., 2016a) (right in Figure 5). We additionally considered $l_2$ attacks, per Carlini and Wagner (Carlini & Wagner, 2017), but find that all of the robust models, whether based on PGD or dual-perturbation attacks, are successful against these.

Our first observation is that *AT-Dual* is significantly more robust to $l_\infty$-bounded PGD attacks than the adversarial training approach in which adversarial examples are generated using $l_2$-bounded PGD attacks (Figure 5 (left)). Consequently, training with dual-perturbation attacks already exhibits better ability to generalize to other attacks compared to conventional adversarial training.

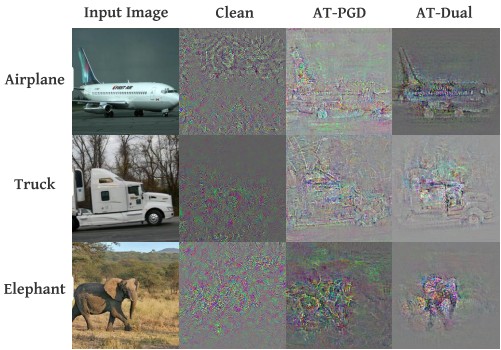

Figure 6: Visualization of loss gradient of different classifiers with respect to pixels of *non-adversarial* inputs.

The gap between dual-perturbation-based adversarial training and standard adversarial training is even more significant when we consider $l_\infty$ dual-perturbation attacks (middle left and middle right figures of Figure 5). Here, we see that robustness of PGD-based adversarially trained model is only marginally better than that of a clean model under large distortions (e.g., when $\epsilon_B \geq 20/255$ in the middle right plot of Figure 5), whereas *AT-Dual* remains relatively robust.

Finally, considering JSMA attacks (see Figure 5 (right)), we can observe that both *AT-Dual* and *AT-PGD* remain relatively robust. However, a deeper look at Figure 5 (right) reveals that compared to *AT-PGD*, *AT-Dual* exhibit moderately higher robustness than the baselines under large distortions of JSMA attacks. Overall, in all of the cases, the model made robust using dual-perturbation attacks remains quite robust even as we evaluate against a different attack, using a different norm.

### 5.5 ANALYSIS OF DEFENSE

Finally, we conduct an exploratory experiment to study adversarial robustness by investigating which pixel-level features are important for different classifiers at prediction time . To do this, we visualize the loss gradient of different classifiers with respect to pixels of the same *non-adversarial* inputs (as introduced in Tsipras et al. (2019)), shown in Figure 6. Our first observation is that the gradients in response to adversarially robust classifiers (AT-PGD and AT-Dual) align well with human perception, while a standard training model (Clean) results in a noisy gradient for the input images. Second, compared to adversarial training with the conventional PGD attack (AT-PGD), the loss gradient of AT-Dual provides significantly better alignment with sharper foreground edges and less noisy background. This indicates that adversarial training with the dual-pertubation attack which models unsuspiciousness can extract more perceptual semantics from an input image and are less dependant on the background at prediction time. In other words, our defense approach can extract highly robust and semantically meaningful features, which contribute to its robustness to a variety of attacks.

## 6 CONCLUSION

In this paper, we proposed the dual-perturbation attack, a novel threat model that produces *unsuspicious adversarial examples* by leveraging the cognitive distinction between image foreground and background. As we have shown, our attack can defeat all state-of-the-art defenses. By contrast, the proposed defense approaches using our attack model can significantly improve robustness against unsuspicious adversarial examples, with relatively small performance degradation on non-adversarial data. In addition, our defense approaches can achieve comparable to, or better robustness than the alternatives in the face of traditional attacks.

Our threat model and defense motivate several new research questions. The first is whether there are more effective methods to identify foreground of images. Second, can we further improve robustness to dual-perturbation attacks? Finally, while we provide the first principled approach for quantifying suspiciousness, there may be effective alternative approaches for doing so.

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
