# OpenReview forum: "Towards Robustness against Unsuspicious Adversarial Examples"
_ICLR.cc/2021/Conference — Reject_

### Official Review · AnonReviewer4 · 2020-10-19
**Maybe we can pick a better term than "suspicious"?**

**Rating:** 4
**Confidence:** 4

**Review:**

The authors present a new adversarial attack, the dual-perturbation attack, that segments the image into 'foreground' and 'background' pixels, and then allows significantly higher perturbation in the 'background' pixels, subject to the (soft) constraint that the "salience" of the image remains high.  When a classifier is trained with dual-perturbation adversarial samples, it seems to be more robust than classifiers trained without dual-perturbation adversarial samples, even when the model is attacked with samples produced with a different attack than the dual-perturbation attack.

However, most of the results (Figures 4 and 5) seem to show that models are much more robust against dual-perturbation attacks when trained with dual-perturbation samples.  I don't find this particularly surprising.  Improvements are marginal---and performance is even degraded for some perturbation values---when compared against non-dual-perturbation attacks for other models (Figure 4, right; Figure 5, right).  The results shown in the appendix, especially for transferability, are similarly marginal (compared against the AT PGD model, which is the only other 'robust' model the authors test against).

So, I am not sure that the results are sufficiently valuable for publication in ICLR.  There are two other primary points that color my review here:

First, the name "suspicious".  To call something "suspicious" is a completely subjective call; when I look at, say, the picture of the truck in Figure 21 in the appendix, I consider the perturbations to the background suspicious.  On the other hand, the perturbations to the stop sign in the referenced Eykholt et al. work are far less suspicious to me, even though the perturbations are large in the foreground.  It is strange, but not suspicious.  I don't want to necessarily get drawn into an argument over what is and what isn't suspicious, but I am sure that we can agree that the definition of suspicious itself is subjective.  Therefore, the idea of an *objective* measure that defines suspiciousness is, for me, a non-starter, and unnecessarily clouds and confuses the issue.  Perhaps the authors could consider the term "salient", but even then that is another subjective term (though perhaps less so).  "Background salience"?  In any case, I can't support the use of the term suspicious as it is currently used.

Secondly, the relation to physical attacks is not clear.  The authors point to physical attacks as both a motivation for studying adversarial samples in general, and also for the distinction between foreground and background.  But in a physical attack, it would be very difficult to modify the background in a sufficiently adversarial way, since, e.g., a stop sign might be viewed from many angles, whereas an attack exploiting the background pixels could (probably!) only work from one very specific angle!  Yet, I can understand that not all attacks are physical, and so my comment here is more about the paper's clarity; I found this point somewhat confusing, and if the authors could do a better job of clarifying their motivation---and why their work is important despite the fact that physical attacks could not in general modify any background pixels.

The paper itself is well-written and clear and I certainly do appreciate the authors' efforts in that regard.

Here are some other minor comments that I hope the authors will find useful:

 - Better clarity on the form of the adversary's utility function in (1) would be helpful.

 - Figure 3 is small and hard to read.  Please find a way to scale it up.

 - Does the definition of relative salience on page 4 need to be normalized to the sum of s_i over the foreground pixels?  (Or over all pixels?)  I think there may be an error here.

Thanks to the authors for the submission.  Despite my negative review, I did find the paper interesting and enjoyable to read.

## Post-rebuttal comments

Thanks to the authors for the comments.  I will leave my review unchanged for this one.

> The term “suspicious”: we agree with the reviewer that the word “suspicious” is an informal and subjective term, and there is no universal notion of what it means for adversarial examples to be unsuspicious. In fact, the core contribution of our paper is to address this issue.

To be clear, what I mean is that you should choose a *different* term than "suspicious".  While it may not sound as good to use a more specific technical term describing your actual measure that you are introducing, it will prevent the inevitable confusion that arises when an informal, vague term is overloaded to have a specific, quantitative definition.

---

> ### Author Response · Authors · 2020-11-25
> **Author Response to Review #4**
>
> The term “suspicious”: we agree with the reviewer that the word “suspicious” is an informal and subjective term, and there is no universal notion of what it means for adversarial examples to be unsuspicious. In fact, the core contribution of our paper is to address this issue. Specifically, we propose a simple formalization of suspiciousness by leveraging cognitive science.
>
> Physical attacks: our attack model is motivated by and similar to physical attacks such that our goal is to make the perturbation large but unsuspicious in the digital space. The notion of physical attacks in our paper is only used to demonstrate that imperceptibility for perturbations is far from necessary,  and we make no claim that our attack method is physically realizable. We will clarify this.
>
> Definition of relative salience: we use the foreground score  (FS) to represent relative salience. Here, FS denotes the fraction (or probability) of human’s attention on the foreground, and it does not need further normalization.

---

### Official Review · AnonReviewer2 · 2020-10-27
**Study of a threat model that decomposes an image into foreground and background with results in-line with expectations**

**Rating:** 6
**Confidence:** 4

**Review:**

Summary
=======
The paper studies threat models for images which take into account the foreground and background within the image. Specifically, they use DeepGaze II to identify pixels as either foreground or background, and use an Lp threat model with a larger radius in the background and a smaller radius in the foreground. The results are largely what one would expect: defenses for different threat models perform poorly against this attack, in comparison to adversarial training on adversarial examples generated by the attack. These attacks have good foreground score, though this is according to the same DeepGaze II model used to generate the examples to begin with.

Overall, my impression is lukewarm. Although there are certainly some aspects which could be improved, there does not appear to be anything glaringly incorrect and the results are in line with expectations. The rest of this review is separated into comments which I would be happy to discuss with the authors, and other minor aspects that the authors can take into account in their revision.

On a general note, I'm not so sure if "unsuspicious" is the right qualifier for the work in this paper. The most relevant work to this paper is that by Xiao et al., however the authors simply give a very brief statement saying that Xiao et al. do not ensure that their attack is unsuspicious. Suspicious is very subjective: a human may very well look at the adversarial example Figure 2 with the corrupted background and think that this looks suspicious, and it's not at all obvious to me why this is any more or less suspicious than the discrete background changes studied by Xiao et al. To be clear, this is not a point of contention (as in this was not incorporated this into my score), but perhaps the authors can take this into account.

Comments for discussion
=======================
1. The datasets considered are quite small (STL-10 and ImageNet-10 are 5k training, 100 test, and Segment-6 is 18k train, 1.2k test), all of which are smaller than even CIFAR10. Is this a limitation of the the DeepGaze model or some other aspect of the approach?

2. In section 3.2, the paper describes how the projection for the PGD attack is a computationally challenging problem since it is not an Lp ball. However, I believe this setting is not any more difficult than the standard PGD attack: since the perturbation is separated into disjoint foreground and background pixels, the projection is exactly equivalent to the standard projection over each set. Upon looking at the algorithm in the Appendix, this is exactly what is done by the authors. The resulting dual-perturbation attack is consequently a natural and straightforward application of PGD to the proposed threat model. This is not a negative aspect; however the text implies that this is splitting and merging is somehow substantially different from PGD when it isn't. Is my understanding here correct?

3. The presentation of the claims of improvement in standard adversarial robust accuracy are somewhat misleading. For example, in Figure 4 the paper shows improved standard robust accuracy past epsilon=2 for the dual training, and the authors attribute this in the text at the end of section 5.3 to their defense, which trains on larger epsilon in the background. This is a misleading comparison because the baseline L2 adversarially trained model is only trained to be robust up to epsilon=2. In order to meaningfully conclude the final sentence of the section, i.e. that AT-Dual can achieve 20% more accuracy at a larger threshold of epsilon=3, it would only be fair to compare to a baseline which has been trained to be adversarially robust up to the same threshold that is being compared, otherwise the new approach will work better simply because the baseline is handicapped by the radius used at training and not necessarily because splitting foreground and backgrounds is any better.

Minor comments
==============
+ In section 2.2, the paper cites a couple of papers for adversarial training as an empirical defense. Among these, Cohen et al 2019 is listed. However, this is not adversarial training; Cohen et al 2019 is a certified defense with guarantees, and is quite distinct from the adversarial training paradigm described in the text.

+ Given the closeness in setting of Xiao et al. 2020 to the work done in this paper, there should be an additional sentence or two in the related work distinguishing the work in this paper from Xiao et al. 2020 other than mentioning this arbitrary concept of "unsupiciousness" of their attacks. In my understanding, the specifics are actually quite different (i.e. the threat model and how to get the backgrounds), but this is not at all indicated in the text which does not give a very meaningful description beyond "suspiciousness" of how these works are actually different.

Update
======
I have read the author response, which largely does not change my score. I would kindly point out to the authors that the "the technical challenge is that the feasible region is not an ball, and computing the projection is challenging in high-dimensional settings" is in fact not very challenging at all. As mentioned in my initial review, the "split and merge" is exactly the standard projection operator on the proposed set and is not a "new heuristic", and so it is exactly PGD and not an adaptation of PGD.

I also understand the other reviewers concerns on suspiciousness, which we all brought up. This likely needs to be thought about and presented more carefully, for example by posing it more formally if the authors insist on this framing.

---

> ### Author Response · Authors · 2020-11-25
> **Author Response to Review #2**
>
> Datasets: we found that existing approaches for salience analysis does not work on small images (e.g., CIFAR-10). Therefore, we choose datasets with higher resolutions. This allows us to appeal to state-of-the-art methods for saliency detection.
>
> Split and merge:  we do not claim that the split/merge method is substantially different from PGD; instead, as we describe in Section 1, our method is an adaptation of the PGD attack. In fact, we can directly use PGD heuristic to solve our optimization problem. However, the technical challenge is that the feasible region is not an $\ell_p$ ball, and computing the projection is challenging in high-dimensional settings. Our split/merge method provides an effective heuristic to address this challenge.

---

### Official Review · AnonReviewer3 · 2020-10-28
**Interesting formalization of unsuspicious adversarial examples, but experiments do not substantiate claims made**

**Rating:** 3
**Confidence:** 5

**Review:**

# Summary of Contributions
The paper proposes a formalization for one aspect of what makes adversarial examples suspicious, positing that visible perturbations to the foreground arouse more suspicion. Based on this proposed threat model the paper proposes the dual-perturbation attack, which perturbs the image with small perturbations in the foreground and larger perturbations in the background, with the background and foreground distinguished either by a provided mask or the DeepGaze II model, while maintaining a low saliency score in the background. The authors successfully generate images with low background saliency, and show that adversarial training with the dual-perturbation attack generates a network that is robust to the dual-perturbation attacks.

# Score Recommendation
I am recommending a rejection for this paper. While the overarching idea of identifying a class of “unsuspicious” adversarial examples is interesting, the formalization itself (which relies on a computation model, DeepGaze II) is not robust to adversarial perturbations. Furthermore, the experiments conducted do not substantiate the claims made (in the last two sentences of the abstract) about the effectiveness against classifiers robust to conventional attacks and the robustness of models trained via adversarial training with dual-perturbation. Finally, the approach does not consistently succeed at generating adversarial examples that are visually unsuspicious.

# Strengths
I appreciated that the paper attempts to introduce a formalization for a new class of attacks going beyond the existing $l_p$ norm-bounded attacks. In addition, the example provided in Figure 1 shows that increasing $\lambda$ _does_ result in perturbations to the background that make it less salient.

# Weaknesses

## Potential for Attacks on Saliency Models
One of my concerns is that the saliency model is designed only for unperturbed images, and thus can itself be attacked. This could lead to cases where the perturbed image is suspicious (specifically, the background is actually salient to humans) but the saliency model has a low score for the saliency of the background. This would compromise one of the key planks of the approach.

## Unconnected Experiments

### Evaluation of attack strength on robust classifiers
The paper demonstrates that the dual-perturbation attack is successful against robust models. However, the fact that the attacks succeed is not really meaningful, as the models attacked are _not_ trained to be robust against the magnitude of attacks applied.

#### AT-PGD network
The AT-PGD network is trained to be robust to attacks with $l_2$-norm up to $\epsilon = 2.0$ (see the fourth paragraph of page 6. As such, it is unsurprising that it would be vulnerable (as shown in the left and middle plots of Figure 4) to attacks where $\epsilon_B > 10$!

#### Randomized Smoothing Networks
- $l_\infty$ attacks: Randomized smoothing is intended to generate models robust only against attacks with bounded $l_2$ norm. The success of $l_\infty$ attacks in Table 5 of Appendix J.1 is thus not relevant.
- $l_2$ attacks: The total attack budget for the smallest non-zero attack presented in Table 5 of in Appendix J.1 is $\sqrt{(\epsilon_F^2 + \epsilon_B^2)} \approx 1.27$, which is >2.5x the attack budget (127/255) designed for in [Certified Adversarial Robustness via Randomized Smoothing](https://arxiv.org/pdf/1902.02918.pdf). It is thus unsurprising that the attack succeeds (merely because it is so large) even if the budget needs to be distributed between the foreground and the background.

#### Suggestion for Improvement
It would be far more interesting to constrain the _total_ budget for the dual-perturbation attack to be equal to that of the PGD attack. For example, in the $l_2$ case, we would have $\sqrt{(\epsilon_F^2 + \epsilon_B^2)} = \epsilon$. This would mean that the allowed perturbations for the dual-perturbation attack are a strict subset of that for the PGD attack. Having a significant proportion of samples vulnerable to PGD attack are also vulnerable to the dual-perturbation attack would be a convincing demonstration that the dual-perturbation attack is a strong attack (specifically, that perturbations primarily in the background are sufficient to change the classification of these images).

### Evaluation of AT-Dual defense against additional attacks
Figure 5 shows that AT-Dual is more robust as compared to AT-PGD on $l_\infty$ PGD attacks and $l_0$ JSMA attacks. This is unsurprising since AT-Dual is trained to be robust against perturbations of larger overall magnitudes (and thus should be expected to be more robust against all attacks in general), albeit at a cost to its clean accuracy (of ~5%; see the right plot in Figure 4 for example).

#### Suggestion for Improvement
It would be more interesting if it could be shown that AT-Dual has a clean and robust accuracy comparable to networks trained to be robust against $l_\infty$ and $l_0$ attacks.

## Failure to generate unsuspicious adversarial examples
One of the focuses of the paper is generating unsuspicious adversarial examples. Unfortunately, limitations of the algorithms computing saliency mean that the proposed attack often fails to achieve this goal; instead adversarial examples generated by the described dual-perturbation attack are often _more_ suspicious. This is the case since the boundaries between the “foreground” and “background” regions are sharp, and is worst when the background is relatively uniform and the saliency algorithm incorrectly identifies part of the background as the foreground. Good examples of suspicious adversarial examples generated by dual-perturbation attack can be seen in the STL-10 and ImageNet-10 images in Appendix L.

# Post-Rebuttal Comments
I'm afraid I'm maintaining my score as a 3.

The authors' response did not do enough to address my concerns:

### Potential for Attacks on Saliency Models
Unless I'm misunderstanding, the perturbed images presented in Fig. 1 are the results of adding perturbations intended to cause a change in the behavior of the **classifier**, not the **saliency model**.

### Evaluation of attack strength on robust classifiers
I still don't understand why the success of the dual-perturbation attack against models that were not trained to be robust to such attacks is of interest.

### Suspiciousness of adversarial examples
The images identified in Appendix L are visually suspicious due to the sharp boundaries between the "foregorund" and "background", not because of a noticeable change to the background. The fact that these images are considered unsuspicious by the metric defined in the paper suggests that the metric needs to be modified.

---

> ### Author Response · Authors · 2020-11-25
> **Author Response to Review #3**
>
> Attack on saliency models: great point! While the saliency model (DeepGazeII) is trained on non-adversarial data, empirically, it exhibits reasonable robustness on perturbed images. As illustrated in Fig.1, the right image shows significantly less background salience compared to the middle right image, and the former has a higher foreground score than the latter.
>
> Evaluation with other attacks: our goal of using alternative attacks is to study the generalizability of the robustness obtained by adversarial training with dual-perturbation examples. However, one can not obtain a generalizable robust classifier by merely and blindly increasing $\epsilon$ of the attack model in adversarial training. In fact, using a large $\epsilon$ can make adversarial training difficult to converge. The key is, in what way should we increase the overall perturbation to improve the generalizability. Our work provides a promising direction to this question. As shown in Fig.6 of the paper, our AT-Dual can extract features that align better with human perception compared to conventional adversarial training.
>
> Suspiciousness of adversarial examples: in our paper, unsuspicious adversarial examples are those whose foreground is significantly more salient than the background. While the proposed dual-perturbation attacks produce unsuspicious examples, their background can still be noticeable.

---

### Official Review · AnonReviewer1 · 2020-10-31
**Review of Reviewer 1**

**Rating:** 4
**Confidence:** 4

**Review:**

I like the idea to check whether it might be possible that noise can be injected selectively in areas where it is likely to make the adversarial example less suspicious and thus it might be possible to inject more noise in those directions. But the challenge is that this might increase the saliency of those areas. The authors propose to use recent work from DeepGazeII to detect saliencies of various pixels.

But I think this poses various problems, which I illustrate as below. Primarily, I am not sure that constructing attacks using  the same model that is then used to measure the saliency of various parts of the image is an unbiased way of doing this.

The Foreground score of $x+\delta$ is defined as  $\mathcal{S})(x+\delta)$ ( using DeepGazeII) (Sec 3.2). Then the mere facts that the adversarial images generated using $S(x+\delta)$ in the adversarial example formulation algorithm has a higher value of Foreground score does answer positively to the question "whether foreground as we identify it is in fact significantly more salient than the background". The algorithm is  maximizing the same function it claims is an impartial judge of saliency. I believe this is a faulty evaluation.

In terms of the main problem looked at in the paper, I am not convinced that it is well-defined. Is the objective to propose a stronger form of attack than existing PGD attack. If that is the case, then the comparison between the attacks need to be fair and I think this point about fairness of comparison need to emphasized as it is slightly confusing.

For example, are the AT-PGD models trained with an $\ell_2$ adversary with $\epsilon=0.2$ whereas the AT-Dual adversaries are trained with the dual-perturbation attack with $\epsilon_F=2.0, \epsilon_B=20.0$? In that case, if $\|\mathcal{F}(x)\|_0=c_1$ and  $\|\mathcal{B}(x)\|_0=c_2$, then the net $\ell_2$ perturbation allowed for the adversary while AT-Dual training is $\sqrt{2c_1}+\sqrt{20c_2} \gg \sqrt{(c_1+c_2)2}$. If I am not missing anything, it looks like one of the adversarial training is considerably stronger than the other and thus comparison between them seems unfair.

In Page 4, the authors mention that satisfying the two hard constraints simultaneously might be difficult. However, the problem is essentially $\ell_2 $ or $\ell_\infty$ balls on some subsets of the pixels. For projection to the $\ell_\infty$ ball, it is enough to make sure that the maximum of the two subsets of pixels are constrained. For the $\ell_2$ case, it is also an intersection of two convex constraints, which should be easily solvable (especially because the two subsets are non-intersecting)


In Fig 5 where the X-axis is $\epsilon$ i.e. where both the AT-PGD models and the AT-Dual models are attacked with the same strength adversary during training, the clean accuracy (i.e. $\epsilon=0$) is quite low for At-Dual model compared to AT-PGD models (and Clean). I think is due to my initial point about the difference in the strength of adversary  used for training the models.


### Update

I have read the author's rebuttal and the comments of the other reviewers. Sadly, the comments of the other reviewers seem to align with my own thoughts and the rebuttal doesn't resolve those questions. I would like to maintain my initial rating of 4.
 More generally using the same model to measure saliency as well as to construct adversarial attacks is not properly justified and secondly, I am not convinced that putting a large perturbation on the "background" as determined as DeepGaze  is a proper side-by-side comparison with AT where AT is allowed  much smaller perturbation.

---

> ### Author Response · Authors · 2020-11-25
> **Author Response to Review #1**
>
> Foreground scores: good point! For any non-adversarial input, we use DeepGazeII to identify its foreground and background. We found that such a partition results in a foreground score (FS) above 0.8 for STL-10, and ~0.85 for Imagenet-10. Here, an FS that is significantly higher than 0.5 indicates that human fixation is relatively localized to the foreground. Thus, our foreground/background partition is reasonable, as the partition aligns with the fact that the foreground is more salient than the background for clean images. Note that when producing adversarial examples, we use the same partition that we derive from clean images. We then use the FS term in the objective function to explicitly control the background salience. As shown in Fig.1, this does help to produce dual-perturbation examples that have less background salience.
>
> Objective and fairness: our goal is not to propose a stronger attack than PGD, though our experiments show this. Our goal is twofold: 1) to model suspiciousness, and 2)  to make use of our suspiciousness modeling to produce adversarial examples and ensure that the produced perturbations do not make the background highly salient.
>
> Computing dual perturbations: the difficulty of solving the optimization problem in Eq.(3) is that the feasible region is not an $\ell_p$ ball, so it requires extra effort to compute the projection, especially in high-dimensional settings. Our split and merge approach provides an effective way to address this challenge by solving two optimization problems, each of which contains an $\ell_p$ ball feasible region and can easily incorporate the existing methods.
>
> Clean accuracy: compared to AT-PGD and clean model, our proposed AT-Dual exhibits similar performance on non-adversarial data, as shown in Fig.5 (left) where $\epsilon=0$.

---

### Author Response · Authors · 2020-11-25
**Response to all: fairness of evluation**

We are grateful to the reviewers for their thoughtful comments. Before providing a point-by-point response to the reviewers’ comments, we wish to clarify the fairness of our evaluation in the paper.

Fairness: when compared to the conventional PGD attack in our experiments (as well as adversarial training methods that rely on these attack models), the proposed dual-perturbation attack uses the same magnitude of foreground perturbations but larger ones on the background. Such comparisons are fair as the two attacks have the same extent of modification on the semantics of input images. Note that the semantics of the object of interest is more likely to be preserved in the foreground; thus, the same magnitude of perturbations on the foreground indicates the same level of modification on the semantics. Moreover, our experiments show that it is unnecessary to make perturbations imperceptible in order to have them unsuspicious.

---

### Decision · Program_Chairs · 2021-01-07
**Final Decision**

**Decision:**

Reject

**Comment:**

The pursued here goal to explore what a broader and more nuanced notion of "imperceptible" perturbation is quite intriguing and could be a basis of really impactful investigations. However, as pointed out in the reviews and comments, the current treatment of this topic suffers from significant presentation and framing shortcomings.

In particular, this work would benefit from stating clearly what is the main topic of study (and what is not), as current framing tends to confuse the readers. It would be also useful to discuss (and acknowledge) that having the split into background and foreground being driven by a (most likely non-robust) model makes the resulting notion of perturbation rather tricky to consistently analyze/certify. Also, the analysis of the properties of the model, although done fairly competently, is missing some important aspects.

Still, once these points are properly addressed, this would constitute a valuable contribution.